# Impact of Global Climate Change on the Distribution Range and Niche Dynamics of *Eleutherodactylus planirostrish* in China

**DOI:** 10.3390/biology11040588

**Published:** 2022-04-13

**Authors:** Chaosheng Mu, Xuecheng Guo, Youhua Chen

**Affiliations:** 1CAS Key Laboratory of Mountain Ecological Restoration and Bioresource Utilization & Ecological Restoration Biodiversity Conservation Key Laboratory of Sichuan Province, Chengdu Institute of Biology, Chinese Academy of Sciences, Chengdu 610041, China; mucs@cib.ac.cn (C.M.); guoxc1@cib.ac.cn (X.G.); 2University of Chinese Academy of Sciences, Beijing 100049, China

**Keywords:** *Eleutherodactylus planirostris*, invasive species, habitat suitability, China, maximum entropy

## Abstract

**Simple Summary:**

*Eleutherodactylus planirostris* has a strong dispersal ability, and the main route of introduction to new regions is likely due to transport via seedlings. This species is taken into account as one of the foremost successful invasive amphibian species with direct or indirect negative impacts in multiple regions. In our study, we predict the potential distribution of *E. planirostris* in China by species distribution models (SDMs) methods. The results show that this species has a much larger suitable habitat area in China than reflected by the current distribution, so the species is likely to spread from the Pearl River Delta to surrounding areas. Under future warming, its invasive range will expand northward in China.

**Abstract:**

Species distribution models (SDMs) have become indispensable tools in risk assessment and conservation decision-making for invasive species. *Eleutherodactylus planirostris* has a strong dispersal ability, and the main route of introduction to new regions is likely transport via seedlings. This species is understood as one of the foremost successful invasive amphibian species with direct or indirect negative impacts in multiple regions. In this study, we used MaxEnt to assess suitable areas for this species under current and future climates globally and in China. We considered seven climatic variables, three timepoints (current, 2050, and 2070), and three CO_2_ emission scenarios. Annual mean temperature, precipitation of the driest month, and annual precipitation were the most important variables predicting *E. planirostris* occurrence. This species has a much larger suitable habitat area in China than reflected by the current distribution, so the species is likely to spread from the Pearl River Delta to surrounding areas. Under future warming, its invasive range will expand northward in China. In conclusion, this study assessed the risk of invasion of this species and made recommendations for management and prevention.

## 1. Introduction

As the global climate warms, the possibility of invasive alien species (IAS) introduction and spread will increase [1,2]. It is predicted that global warming will change the geographical range of species [3] and that the viability of species in a changing climate will reflect their dispersal capacity [4]. Species with a narrow niche and poor dispersal capacity, such as many endangered species, are more sensitive to these changes [5] than species with good dispersal capacity [6], such as IAS. Most IAS are better adapted to local conditions, beating slow-spreading species and eventually driving them to extinction [7].

Global warming is affecting China’s ecosystems, as is happening in other parts of the world. [8]. Global climate change has shown a huge influence on species distributions [9,10,11,12,13]. For instance, global warming has altered the distributions of butterflies [14], birds [15], amphibians [16], and mammals [17]. The negative effects of global climate change on ecosystem function have attracted the attention of governments and scientists all over the world [18,19]. China is greatly rich in species and is listed as one of the 12 countries in the world with great biodiversity, also known as a “megadiversity country” [20]. Chinese diverse habitats and climatic conditions make it particularly vulnerable to the settlement of IAS.

The greenhouse frog *Eleutherodactylus planirostris* [21] originated on islands in the Caribbean Sea area [22] and has now invaded the southeast United States, some Pacific islands [23,24], and Hong Kong, China [25]. The species has a strong dispersal ability, and the main route of introduction is likely transport via seedlings [26,27]. This species is considered one of the most successful invasive amphibian species [28], having direct or indirect negative impacts in multiple regions [24,29]. *E. planirostris* expanded rapidly after invading Hawaii. The current density has been found to be as high as 12,500 individuals/hm^2^, with this population consuming 129,000 individuals/hm^2^ of invertebrates every night [24].

In this study, grounded on available occurrence records of *E. planirostris* and high-resolution environmental data on climate warming, we modeled the potential global distribution of *E. planirostris*. We are particularly curious about the potential invasion dynamics of the species in China. The purposes of this study are to (1) identify key environmental variables that are highly correlated with the current range of *E. planirostris* and (2) to describe the current potential distribution and to model its distribution under future climate change scenarios to help control the invasion of the species in China.

## 2. Materials and Methods

### 2.1. Species and Environmental Data

We obtained occurrence records for *E. planirostris* from the Global Biodiversity Information Facility (GBIF, obtained on 11 January 2022), and those occurrence records were removed if it is outside the environmental data. To reduce spatial autocorrelation, to make sure there was only one occurrence record in each grid cell by the ENMTools R package [30]. In the end, 1450 occurrence records of *E. planirostris* were obtained for this study.

A total of 19 global bioclimatic variables were acquired from the WorldClim database. Future climate scenarios data were accessed from the IPCC’s 5th Assessment Report. The IPCC coordinates climate research communities to develop a suite of situations that mirror attainable climate scenarios for the 21st century. The “Representative Concentration Pathways” (RCPs) describe assumptions about possible future concentrations of greenhouse gases [31]. In this research, we used the Global climate model of the Beijing Climate Center (BCC-CSM1-1), because we are mainly concerned about the invasion of this species in China. The BCC-CSM1-1 is one of the most commonly used global climate models [32]. We chose RCP 2.6 as a stringent mitigation scenario, RCP 6.0 as a general mitigation scenario, and RCP 8.5 as a scenario without additional efforts. All environmental data were obtained with a high resolution of 2.5 arcmins (5 km × 5 km). We used Pearson’s rank correlation to examine the cross-correlation (|r| > 0.70) and removed highly correlated variables to avoid collinearity in statistical models [33]. The final climate variables in our modeling and analysis include Bio1 = Annual Mean Temperature, Bio2 = Mean Diurnal Range, Bio3 = Isothermality, Bio10 = Mean Temperature of Warmest Quarter, Bio12 = Annual Precipitation, Bio14 = Precipitation of Driest Month, and Bio18 = Precipitation of Warmest Quarter.

### 2.2. MaxEnt Model

MaxEnt is a sophisticated machine learning method based on the maximum entropy model, which can predict species distribution from data on species occurrence and environmental variables [34,35]. The calibration phase is extremely vital for the rigorous construction of the model, and its goal is to work out which combination of parameters can best represent the phenomenon by finding the best match to the data [36,37]. We used the kuenm R package [38] to test candidate solutions, including all 31 possible combinations of 5 feature classes (linear = l, quadratic = q, product = p, threshold = t, and hinge = h) and 10 regularization multiplier settings (0.1, 0.3, 0.6, 0.9, 1, 2, 3, 4, 5, and 6). Partial receiver operating characteristic (ROC) analysis based on 500 iterations, 50% data for bootstrapping, 5% missing rate, and modified Akaike Information Criterion (AICc). The best candidate models were selected according to the following criteria: (1) significant models with (2) omission rates ≤ 5%. Then, from these candidate models, models with delta AICc values ≤ 2 were chosen as final models for mapping and projection [38]. We set 70% of the occurrence points for use in model calibration and the remaining 30% to evaluate the model predictions, with a logistic output format ranging from 0 (unsuitable environmental conditions) to 1 (optimal). The maximum number of background points was 10,000. To make sure the soundness of model predictions, the process ran a bootstrap replicated run type 10 times. The other parameters of the MaxEnt model are set to default values. The creation of ultimate models in MaxEnt and transformations to future conditions can be performed using the parameters selected during calibration. Under free extrapolation settings, responses in areas that are environmentally different from the calibrating area follow trends in the environmental data used for calibration.

The performance was assessed using the area under the receiver-operating characteristic curve (AUC), the value of which increases with an increasing deviation of the species distribution from the random [39]. The evaluation standard was as follows: AUC > 0.9 was regarded as excellent, 0.7 < AUC < 0.9 was regarded as good, 0.5 < AUC < 0.7 was regarded as acceptable, and AUC < 0.5 was regarded as invalid. To understand the importance of environmental variables on species distribution, percentage contribution (PC) was used as an evaluation indicator for MaxEnt. PC is defined as the increase in the contribution of the dependent variable to the regularization gain when the training algorithm is run [40].

### 2.3. Classification of Suitable Habitats

We imported the results generated by MaxEnt software 3.3.4 (AMNH, New York, NY, USA) into ArcGIS 10.2 (ESRI, Redlands, CA, USA). After converting the results into raster format, we reclassified the suitable habitats for *E. planirostris* with thresholds in ArcGIS and divided the suitable environmental conditions into 4 levels based on logistic value; areas were classified as follows: an area unsuitable for *E. planirostris* (0 ≤ *p* ≤ 0.1), low-suitability area (0.1 < *p* ≤ 0.3), medium-suitability area (0.3 < *p* ≤ 0.5), and high-suitability area (0.5 < *p* ≤ 1) [41]. Here, *p* is the predicted probability of occurrence.

### 2.4. Changes in Suitable Habitat Area and Centroids

In this study, we use the 10th percentile training presence threshold to aid explain the model better; the average logical output map of the SDMs is transformed into a binary map of fit (1) and unfit (0) conditions. This threshold rule can be interpreted as a threshold describing the “core distribution” by ignoring the 10% least favorable training times [42]. To further examine trends, binary rasters were used to analyze the predicted contraction, expansion, areas of no change and no occupancy, and centroid changes for *E. planirostris* in China using the Python-based GIS toolkit SDMtoolbox [43].

## 3. Results

### 3.1. Model Selection and Accuracy Evaluation

Among 310 candidate models, only one model parameter combination meets our criteria. In this candidate model (regularization multiplier = 0.9, feature class combination = qt), the mean AUC ratio was 1.927, the partial ROC was 0, the omission rate was 0.05, and the AICc was 36,766.001, which represented the lowest delta AICc after adjustment. The average value of the training AUC for the replicate runs was evaluated as 0.932. The simulation results indicated that the MaxEnt model output provided satisfactory results.

### 3.2. Prediction of Current Distribution of E. planirostris

On a global scale, our results showed that the suitable habitat areas for *E. planirostris* are primarily distributed in southeastern North America, the Caribbean Islands, central and southern South America, a few coastal areas of Africa, some coastal areas and islands of Southeast Asia and Oceania, and southeast China (Figure 1). In China, the habitat suitable for *E. planirostris* was found to be mainly distributed in the southern mainland, Taiwan and Hainan Island. Furthermore, the highly suitable habitat occupied most of these areas (Figure 1). The importance of the environment can be evaluated by the PC value of the environment variable. The three most important bioclimatic variables and their percent contributions were annual mean temperature (Bio1, 27.2%), precipitation in the driest month (Bio14, 23%), and annual precipitation (Bio12, 21.3%), and the cumulative value of the three items reached 71.5% (Table 1).

### 3.3. Prediction of Future E. planirostris Distribution in China

Under different future climate conditions, the potential geographical distribution areas of the three grades are significantly different, but the total suitable area was always much greater than that of current times (Figure 2). Currently, over 10% of climate suitability was found to account for approximately 6.11% of the total study area. As shown, the area of suitable habitat was projected to increase by 77.44% (RCP2.6), 76.35% (RCP4.5), and 114.48% (RCP8.5) in the 2050s (Figure 2). Similar range expansion patterns were observed for the 2070s: 71.95% (RCP2.6), 86.96% (RCP4.6), and 110.27% (RCP8.5). Furthermore, the centroid was predicted to move away from the south (eastern Guangxi Province) to the north (central Hubei Province), particularly under the emission scenario RCP8.5 (Figure 3).

## 4. Discussion

Because risk maps visually describe where IAS are likely to be established, they can be precious tools for a strategic IAS regulation programme. The study suggests that global climate change will benefit the spread of *E. planirostris* in China, thus having great potential to threaten the native ecological system.

We used data from around the globe to describe the climatic conditions of *E. planirostris*. Using only occurrence information from the native or invasive range can seriously underrate potential distribution areas, especially for IAS, whose human factors spread can be very important [44]. Indeed, there are very few occurrences in China [25,45]. However, our results showed that *E. planirostris* has a much larger suitable habitat area in China than expected. The species is likely to spread from the Pearl River Delta to surrounding areas.

In its native place, *E. planirostris* is a common creature that has adapted well to a variety of habitats, including gardens and houses around human activity, forests, mountains, and rivers [46]. In Jamaica, it is often found in relatively dry habitats such as open grasslands, scrub, and roadsides [47]. In Hong Kong, *E. planirostris* often inhabit the forest floor in secondary forests and forest margins [25]. Although it can live in a wide variety of habitats and does not need to live directly in or near water, precipitation remains important for its survival. This might be because overcast or rainy climate conditions are key factors in croak behavior [23], and since frogs are not amniotic animals, humidity is critical for fertilized egg development and hatching success [48].

Studies of the climate requirements of *E. planirostris* have shown that, mainly, the frog has established stable populations outside its native range with climate conditions similar to those in Cuba [28,49]. However, it is also found in the southeastern United States, where seasonal daily minimum temperatures are as low as 4 °C to 8 °C [22], and some studies suggest that prolonged residence in the Florida Keys may have caused frogs to have evolved physiological and/or behavioral adaptations to deal with cooler temperatures [49]. *E. planirostris* populations established in China live in temperature conditions similar to those in the United States. Our findings further demonstrate that *E. planirostris* has adapted to temperatures below those of its native habitat, even though annual mean temperature remains the most important factor limiting *E. planirostris* distribution (Table 1).

The species has been found several times in Hong Kong and Shenzhen in China in the past 10 years, indicating that it has been repeatedly introduced into China and has produced stable reproductive populations [25,45]. The results presented here show that *E. planirostris* has a much larger area of suitable habitat in China than they currently occupy, so the species is likely to spread from the Pearl River Delta to surrounding areas. Relevant departments need to devote attention to this situation to prevent this scenario from happening.

Climate change will allow *E. planirostris* to invade China more effectively. From the present time to the 2070s, the total area of suitable habitat for this species will be greatly expanded with the greenhouse gas emissions continuing to increase. The reason for the expansion of the distribution area in the future is that under the influence of the greenhouse effect caused by the increase in CO_2_ concentration in the atmosphere [41]. Changes in the centroid of the *E. planirostris* suitable area would reflect how far and in which direction the distribution will shift with global climate change scenarios. The centroid of the entire suitable area of *E. planirostris* showed a northward migration trend for one to four latitudes under three different greenhouse gas emission scenarios (the 2050s and 2070s). As a result, the species could threaten to establish further north in the future. Adopting a low-carbon sustainable development path is the most effective way to mitigate the possibility of *E. planirostris* invasion and expansion. On the other hand, the suitable areas of *E. planirostris* will expand northward migration with the development and consumption of fossil fuels. Therefore, sustainable low-carbon development will limit the colonization and spread of IAS and protect our environment [50].

According to the “three-stage hierarchical approach” set by the Convention on Biological Diversity [51], emphasis must be placed on prevention in areas of high suitability where no distribution of the species has been recorded. That means we need to focus on long-term surveillance of areas surrounding successful intrusions and establish rapid response protocols for early detection. In areas where *E. planirostris* has already been invaded, the focus should be on minimizing the negative impact of the species and stopping future spread [52]. Some control options for *E. planirostris* have been developed for other frogs. For instance, chemical controls are used to control *Eleutherodactylus coqui* in Hawaii and are equally effective against *E. planirostris* [53]. High-temperature spray or steam treatments are usually used for various pests carried by plants. High-temperature sprays or steam sprayed on plants at either 45 °C for 1 min or 39 °C for 5 min was effective at eradicating adult E. coqui [54], and similar results are expected for *E. planirostris*, considering their similar thermal tolerances [55]. Maintaining long-term monitoring of discovered populations and posting signs to warn people of the animals’ dangers may be the simplest and most effective ways.

## 5. Conclusions

Because risk maps visually describe where IAS are likely to be established, they can be precious tools for a strategic IAS regulation programme. Although there are very few occurrences in China, *E. planirostris* has a much larger suitable habitat area in China than expected. *E. planirostris* has adapted to temperatures below those of its native habitat. The species is likely to spread from the Pearl River Delta to surrounding areas. Climate change will allow *E. planirostris* to invade China more effectively. Relevant government departments need to focus on long-term surveillance of areas surrounding successful intrusions and establish rapid response protocols for early detection.

## Figures and Tables

**Figure 1 biology-11-00588-f001:**
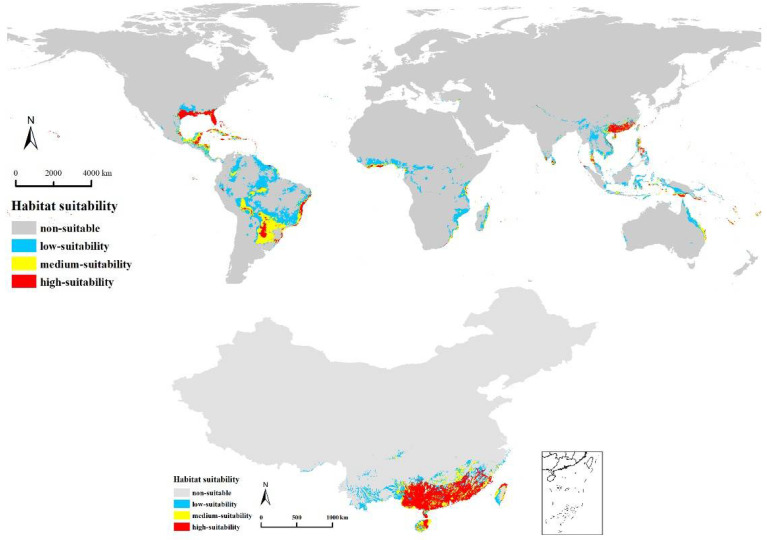
Simulated current environmentally suitable range for *E. planirostris* in China and worldwide.

**Figure 2 biology-11-00588-f002:**
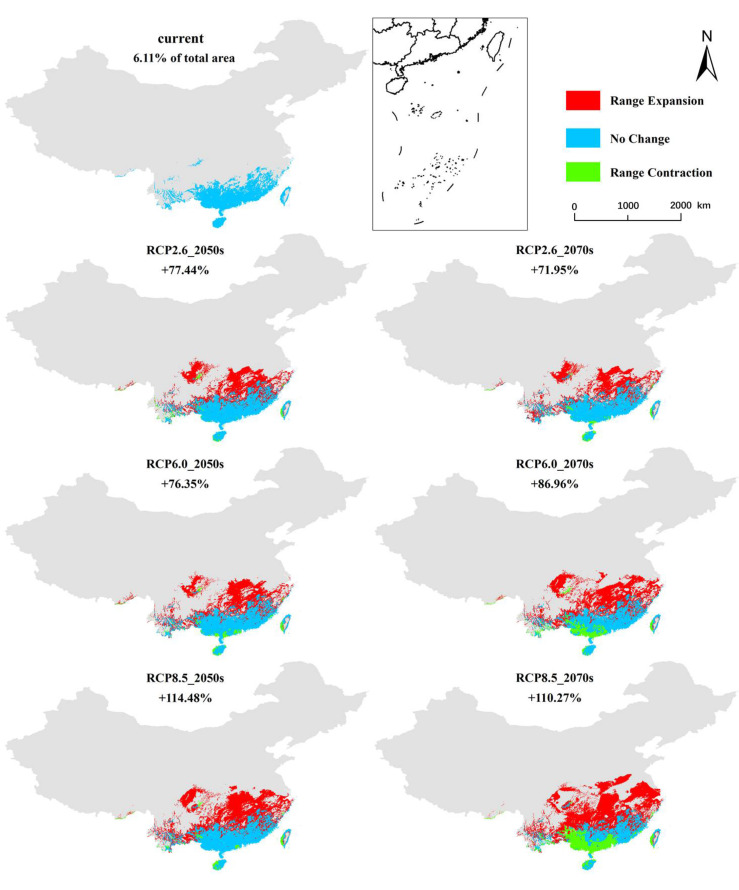
The current and future potential geographical distributions of *E. planirostris* in China from 2050s to 2070s according to the climate scenarios RCP 2.6, RCP6.0, and RCP 8.5.

**Figure 3 biology-11-00588-f003:**
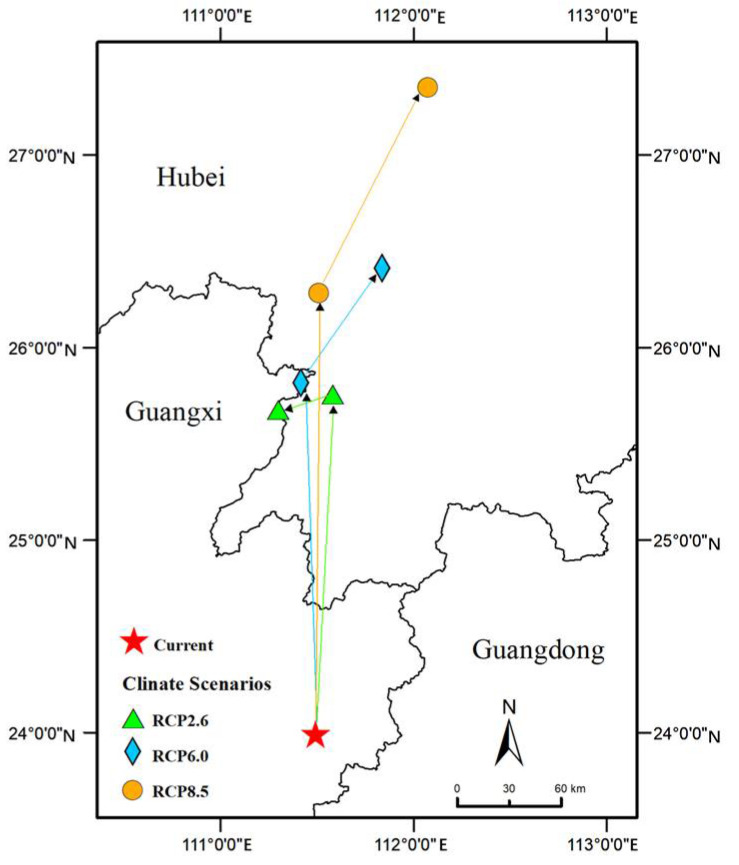
Shift distances of the centroid for *E. planirostris* under three different climate scenarios (RCP2.6, RCP6.0, and RCP8.5) in the future.

**Table 1 biology-11-00588-t001:** Contributions of the environmental predictors influencing the potential distribution of *E. planirostris*.

Environmental Variable	Percentage Contribution (%)
Annual Mean Temperature	27.2
Precipitation of Driest Month	23
Annual Precipitation	21.3
Precipitation of Warmest Quarter	12.4
Isothermality	7.4
Mean Temperature of Warmest Quarter	7.2
Mean Diurnal Range	1.5

## Data Availability

The data presented in this study are available on request from the corresponding author.

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
