# Peer review of "Impact of Global Climate Change on the Distribution Range and Niche Dynamics of Eleutherodactylus planirostrish in China"

_biology, 2022, doi:10.3390/biology11040588_

Round 1
Reviewer 1 Report
I enjoyed reading your paper very much and I think the findings you present will be valuable to both researchers and managers.
I do not have any substantive suggestions for changes. I have made a few suggestions for grammatical changes in the file that is attached. I will also suggest that the editors conduct a minor review for English usage. Thank you.

Author Response
Thank you for providing us with the opportunity along with helpful suggestions to further improve our paper. Following the constructive comments of you and two anonymous reviewers, we believe the revision has become more readable and understandable for the audience.
Sincerely yours,
Authors
Point 1: Suggest saying United States rather than America.
Response 1: thanks for your comment. We have changed “America” to “United States”.
Point 2: The word expanded suggests increasing in the area occupied. Is this what you mean?
Response 2: thanks for your comment. Yes it is.
Point 3: Use past tense.“… sure there is was only one…”
Response 3: thanks for your comment. We have changed “is” to “was”.
Point 4: Suggest beginning this sentence with“We imported the results…”
Response 4: thanks for your comment. We have changed “Import” to “We imported”.
Point 5: Please change the distribution from“southwestern” to“southeast” North America. I would also suggest including the Caribbean Islands
Response 5: thanks for your comment. We have changed “southwestern” to“southeast” and including the Caribbean Islands.
Point 6: The outline drawing in this figure is not described.
Response 6: thanks for your comment. We have revised The map note of Figure 2 as "The current and future potential geographical review of E. Planirostris in China from The 2050s to 2070s According to the Climate Scenarios RCP 2.6, RCP6.0, and RCP 8.5"
Point 7: Delete the word“is” and state“…precipitation remains important…”.
Response 7: thanks for your comment. We have removed the word "is".
Reviewer 2 Report
I have to congratulate the authors of this work because it seems to me a very well conducted and presented simulation research. The results are important for the geographical area in question, but also for a general vision of this type of scenario that can be generated by introductions of alien species, in this case amphibians, also in relation to climatic changes.
Author Response
Thank you for providing us with the opportunity along with helpful suggestions to further improve our paper. Following the constructive comments of you and two anonymous reviewers, we believe the revision has become more readable and understandable for the audience.
Sincerely yours,
Authors
Point : The results are important for the geographical area in question, but also for a general vision of this type of scenario that can be generated by introductions of alien species, in this case amphibians, also in relation to climatic changes.
Response:thank you for your suggestion. The introduction of invasive alien species is indeed widely affected by climate change, so our article simulates future climate change and discusses the risk of amphibian invasion. In the future, our research will expand to other groups of vertebrates such as reptiles, birds, or invertebrates such as insects.
Reviewer 3 Report
This is a straightforward study that was well done. The ms requires more editing for language and grammar, but the majority of it is clean.
On line 149 I believe the authors meant to write "southeastern" instead of "southwestern" North America.
Author Response
Thank you for providing us with the opportunity along with helpful suggestions to further improve our paper. Following the constructive comments of you and two anonymous reviewers, we believe the revision has become more readable and understandable for the audience.
Sincerely yours,
Authors
Point 1: On line 149 I believe the authors meant to write "southeastern" instead of "southwestern" North America.
Response:thanks for your comment. We have changed “southwestern” to “southeastern”